# Exercise-induced enhancement of synaptic function triggered by the inverse BAR protein, Mtss1L

Christina Chatzi[1†], Yingyu Zhang[1†], Wiiliam D Hendricks[1,2], Yang Chen[1,2], Eric Schnell[3,4], Richard H Goodman[1], Gary L Westbrook[1*]

[1]Vollum Institute, Oregon Health & Science University, Portland, United States; [2]Neuroscience Graduate Program, Vollum Institute, Oregon Health & Science University, Portland, United States; [3]Department of Anesthesiology and Perioperative Medicine, Oregon Health & Science University, Portland, United States; [4]Portland VA Health Care System, Portland, United States

**Abstract** Exercise is a potent enhancer of learning and memory, yet we know little of the underlying mechanisms that likely include alterations in synaptic efficacy in the hippocampus. To address this issue, we exposed mice to a single episode of voluntary exercise, and permanently marked activated mature hippocampal dentate granule cells using conditional Fos-TRAP mice. Exercise-activated neurons (Fos-TRAPed) showed an input-selective increase in dendritic spines and excitatory postsynaptic currents at 3 days post-exercise, indicative of exercise-induced structural plasticity. Laser-capture microdissection and RNASeq of activated neurons revealed that the most highly induced transcript was *Mtss1L*, a little-studied I-BAR domain-containing gene, which we hypothesized could be involved in membrane curvature and dendritic spine formation. shRNA-mediated *Mtss1L* knockdown in vivo prevented the exercise-induced increases in spines and excitatory postsynaptic currents. Our results link short-term effects of exercise to activity-dependent expression of Mtss1L, which we propose as a novel effector of activity-dependent rearrangement of synapses.
DOI: https://doi.org/10.7554/eLife.45920.001

**\*For correspondence:**
westbroo@ohsu.edu

[†]These authors contributed equally to this work

## Introduction

The beneficial cognitive effects of physical exercise cross the lifespan as well as disease boundaries (*Mandolesi et al., 2018*; *Saraulli et al., 2017*). Exercise alters neural activity in local hippocampal circuits, presumably by enhancing learning and memory through short- and long-term changes in synaptic plasticity (*Vivar et al., 2013*; *Voss et al., 2013*). The dentate gyrus is uniquely important in learning and memory, acting as an input stage for encoding contextual and spatial information from multiple brain regions. This circuit is well suited to its biological function because of its sparse coding design, with only a few dentate granule cells active at any one time (*Jung and McNaughton, 1993*; *Leutgeb et al., 2007*; *Severa et al., 2017*). These properties also provide an ideal network to investigate how exercise-induced changes in activity-dependent gene expression affect hippocampal structural and synaptic plasticity in vivo.

Most studies have focused on the cognitive effects of sustained exercise (*Bolz et al., 2015*; *Creer et al., 2010*; *Uda et al., 2006*). However, memory improvements are well documented after acute bouts of exercise (*Mandolesi et al., 2018*; *Perini et al., 2016*). Thus, the response to a single episode of exercise may be better suited to uncover cellular and molecular cascades that form and rearrange synapses, and which evolve over minutes to weeks following exercise. We reasoned that these mechanisms could be unmasked by a single period of voluntary exercise in vivo, followed by

functional and molecular analysis of subsequent changes specific to neurons that were activated during the exercise.

## Results

cFos$^{creERT2}$ transgenic mice (Fos-TRAP) provide valid proxies of neural activity (*12* and *Figure 1—figure supplement 1*) and a means to permanently label activated dentate granule cells. During a two-hour exposure to running wheels, mice ran approximately 3 km. We examined activated cells 3 days post-running in Fos-TRAP mice crossed with a TdT reporter line (*Figure 1A*). We used Fos immunohistochemistry at 1 hr post-exercise, confirming robust stimulation of neuronal activity in mature granule cells (*Figure 1B*). The increase in Fos expression, assessed by immunohistochemistry, matched the increase in TdTomato-positive cells (TdT$^+$) measured 3 or 7 day later in Fos-TRAP mice, indicating that activated granule cells were accurately and permanently labeled during the 2 hr time window *Figure 1C*). We refer to these cells as 'exercise-TRAPed'. To investigate whether a single bout of exercise activated a specific subset of granule cells, we exercise TRAPed dentate granule cells (TdT$^+$) and compared this population to an ensemble activated by a subsequent re-exposure to exercise either 1 or 4 days later, as measured by Fos immunohistochemistry at 2 hr after the 2nd exercise period (*Figure 1D*). When the two exercise periods were separated by 24 hr, only 13% of exercise TRAPed cells were activated in the 2nd exercise period. There was almost no overlap between the two neuronal ensembles (1%) when the two periods were separated by 4 days (*Figure 1D*, right panel). Exercise-TRAPed neurons were distributed through the granule cell body layer, without labeling in the subgranular zone. This indicates that our exercise protocol activated stochastic, non-overlapping sets of mature granule cells, consistent with the sparse coding design of this circuit.

Higher cortical information converges on the dentate gyrus through laminated perforant path axons from the entorhinal cortex. To examine whether a single exposure to exercise altered structural plasticity in exercise-activated neurons, we analyzed exercise-TRAPed cells in mice 3 or 7 days post-exercise as compared to homecage controls (*Figure 2A*, left panels). Exercise-TRAPed cells showed a nearly 50% increase in dendritic spines at 3 days. This increase was limited to the outer molecular layer (OML), which interestingly, receives contextual and time information via the lateral perforant path (*Hargreaves et al., 2005*; *Yoganarasimha et al., 2011*), whereas there was no change in dendritic spines in the middle molecular layer (MML), which receives spatial information via the medial perforant path (*Hafting et al., 2005*; *Sargolini et al., 2006*). Consistent with a transient response to a single stimulus, spine density in the OML returned to baseline levels by 7 days (*Figure 2A*, right panels). Exercise did not affect total dendritic length in OML or MML indicating that the increase in spine density reflected an increase in the number of synapses (*Figure 2—figure supplement 1*).

The increase in spine density in the OML was associated with a corresponding increase in functional synaptic input in exercise-TRAPed cells (*Figure 2B*). Three days after exercise, we used acute brain slices to make simultaneous whole-cell voltage clamp recordings from an exercise-TRAPed granule cell (yellow, *Figure 2B*, middle) and a neighboring control granule cell (green, *Figure 2B*, middle). Selective stimulation of lateral perforant path (LPP) axons evoked nearly three-fold larger EPSCs in the exercise-TRAPed cell of the simultaneously recorded cell pair (*Figure 2C,D*), whereas EPSC amplitudes evoked by stimulation of medial perforant path axons (MPP) were unaffected (*Figure 2C,D*). Plotting each cell pair as a function of the site of stimulation indicated that MPP cell pairs were on the unity line, whereas LPP cell pairs were below the unity line, indicative of the increased EPSCs in exercise-TRAPed cells (*Figure 2E*). The increased EPSC amplitudes could not be attributed to differences in intrinsic membrane properties or presynaptic release probability (*Figure 2—figure supplement 2* and *Supplementary file 1*). These results indicate that dentate granule cells activated by a single bout of exercise show a laminar-specific increase in dendritic spines and in excitatory postsynaptic currents.

Activity-dependent gene expression is one of the main drivers of short- and long-term changes in synaptic function (*West and Greenberg, 2011*). To identify exercise-activated genes underlying the observed structural plasticity in the dentate gyrus, we isolated RNA from RFP$^+$ nuclei of exercise-TRAPed cells (*Figure 3A,B*) using laser microdissection. An ensemble of 150 individual exercise-TRAPed or adjacent non-activated (RFP$^-$) mature granule cells was excised per mouse at 3 and 7

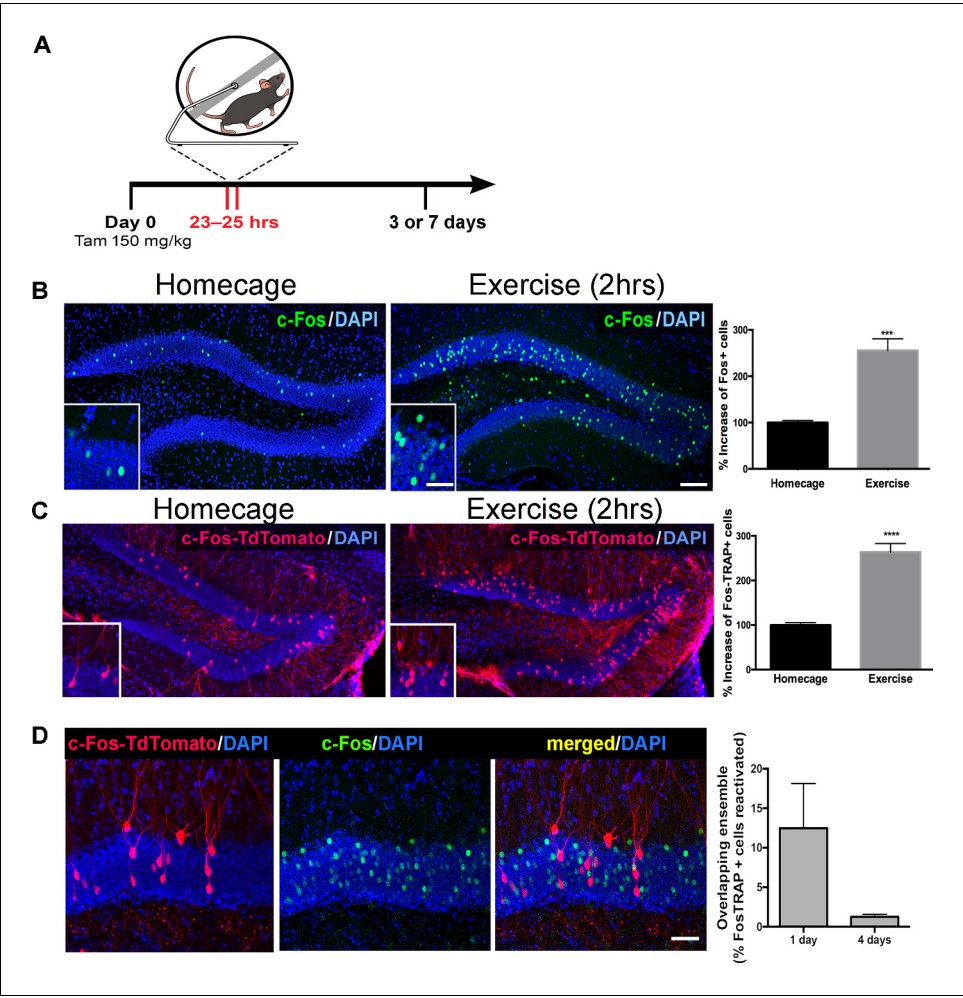

**Figure 1.** Single exposure to running wheel induces transient synaptic plasticity in exercise-TRAPed dentate granule cells. (**A**) Schematic showing exercise paradigm. Fos-TRAP:TdTomato (*Guenthner et al., 2013*) mice were injected with tamoxifen (150 mg/kg) 24 hr before exposure to 2 hr of voluntary exercise, while littermate controls remained in their homecage. Mice were sacrificed 3 or 7 days after exposure to the running wheel. (**B**) Voluntary exercise (2 hr) increased neuronal activity in the dentate gyrus. Representative images of endogenous c-Fos expression in the dentate gyrus of WT mice housed in their homecage (left) or 2 hr after exposure to voluntary exercise (middle). A single bout of exposure to exercise increased c-Fos+ cells in the dentate gyrus (right). (% increase, Homecage: 100 ± 4 n = 6, Exercise: 255 ± 25, n = 4, unpaired t-test p=0.001). Scale bars: insert 50 μm, right 100 μm. (**C**) Representative images of the dentate gyrus from Fos-TRAP:TdTomato mice housed in their homecage (left) or 3 days after 2 hr of voluntary exercise (middle). Voluntary exercise increased exercise-TRAPed dentate granule cells (% increase from baseline in exercise-TRAPed cells, homecage 100 ± 5 n = 5, Exercise 264 ± 19, n = 5, unpaired t-test, p<0.0001). (**D**) A single exposure to exercise tags distinct populations of activated DG granule cells. (**A**) We used the Fos-TRAP: Tdtomato mice to tag a neuronal ensemble activated by a single exposure to exercise (2 hr) (Tdtomato[+]). We compared these exercise-TRAPed cells to granule cells activated by a second exposure to exercise (**C**) and tagged at 2 hr post-exercise using c-Fos immunohistochemistry 1 or 4 days later. Fos-TRAP:Tdtomato mice were injected with Tamoxifen (150 mg/kg) 24 hr prior to exercise. Animals were exposed to a second bout of exercise either 1 or 4 days later. (**B**) When the two exercise periods were separated by 24 hr, 12.5 ± 5.6%, (n = 3) of the exercise-TRAPed cells were re-activated, whereas with a 4-day separation only 1.3 ± 0.3% (n = 4, unpaired t-test, p=0.07) overlapped, indicating that labeling with exercise was stochastic.

DOI: https://doi.org/10.7554/eLife.45920.002

The following figure supplement is available for figure 1:

**Figure supplement 1.** In vitro validation of Fos-TRAP method.

DOI: https://doi.org/10.7554/eLife.45920.003

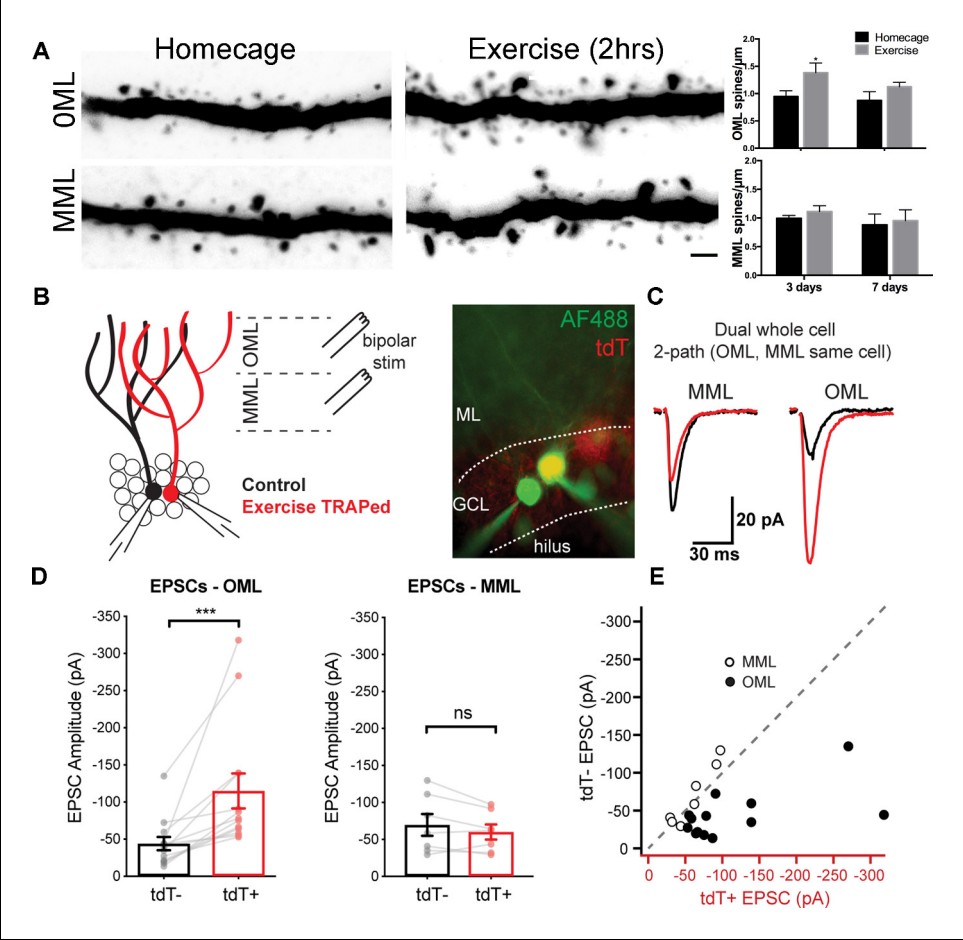

**Figure 2.** A single bout of exercise induces a laminar-specific increase in excitatory synaptic innervation. (**A**) Representative images of TRAPed granule cell dendrites in the OML and MML from mice at baseline (i.e. homecage) or 3 days post-exercise. Scale bar: 5 μm. Spine densities were significantly increased in the OML of activated cells at 3 days post exercise (two-way ANOVA, OML 3 days homecage: 0.94 ± 0.1, exercise: 1.39 ± 0.2, p=0.02, n = 5), whereas there was no difference in the OML at 7 days post-exercise (OML 7 days homecage: 0.87 ± 0.2, exercise: 1.10 ± 0.1, p=0.18, n = 5). Spine density in the MML was unaffected (two-way ANOVA, no interaction between homecage and exercise groups, MML 3 days homecage: 0.99 ± 0.1, exercise: 1.10 ± 0.1; MML 7 days homecage: 0.88 ± 0.2, exercise 0.95 ± 0.2, n = 5, p=0.79). (**B**) Configuration for simultaneous whole-cell voltage clamp recordings of control and exercise-TRAPed dentate granule cells (left). Bipolar stimulating electrodes were placed in the OML and/or MML to activate lateral and medial perforant path axons, respectively. Cells were filled with AlexaFluor 488 dye (right) to confirm that the adjacent cell bodies had dendritic arbors that overlapped (control:green, exercise TRAPed: yellow). (**C**) Representative EPSCs from the cell pair shown at left in response to alternating MML and OML stimulation in the control (black traces) and exercise-TRAPed cell (red traces). (**D**) OML stimulation produced EPSCs that were significantly greater in exercise-TRAPed cells (EPSCs - OML amplitude: TdT⁻, −44.1 ± 8.9 pA; TdT⁺, −114.9 ± 23.5 pA; n = 13 cell pairs (eight mice); p=0.0002, Wilcoxon matched-pairs signed rank test), whereas MML stimulation did not differ from controls (EPSCs - MML amplitude: TdT⁻ −69.7 ± 14.9 pA; TdT⁺, −60.1 ± 10.3 pA; n = 7 cell pairs (four mice); p=0.15, paired t-test). There was no difference in EPSC amplitudes during OML stimulation while recording from two control (both TdT⁻) granule cells (EPSC amplitudes: TdT⁻, −55.8 ± 16.3 pA; TdT⁻, −57.5 ± 13.8 pA, n = 5 cell pairs, p=0.80, paired t-test). (**E**) EPSC amplitudes from control (y-axis) and exercise TRAPed (x-axis) from each cell pair were plotted with each point representing a cell pair. MML stimulated cell pairs (open circles) were present along the unity line (dashed line), whereas OML stimulated pairs (black circles) were shifted below unity, indicating that the larger EPSC amplitudes for exercise-TRAPed cells was specific to the OML.

DOI: https://doi.org/10.7554/eLife.45920.004

The following figure supplements are available for figure 2:

*Figure 2 continued*

**Figure supplement 1.** Dendritic lengths were equal in outer and middle molecular layers of exercise TRAPed cells.

DOI: https://doi.org/10.7554/eLife.45920.005

**Figure supplement 2.** Paired-pulse ratio (PPR) in control and exercise-TRAPed granule cell paired recordings.

DOI: https://doi.org/10.7554/eLife.45920.006

days post-exercise. Analysis of transcripts by RNASeq (*Figure 3—figure supplement 1*) revealed up- and down-regulated transcripts (*Figure 3C*, left panel, *Supplementary file 2* and *3*) of which approximately 150 transcripts were significantly upregulated in exercise-TRAPed neurons 3 days after exercise. Gene Ontology analysis indicated that most of the upregulated transcripts had signaling or synaptic functions. The top ten upregulated genes (*Figure 3C*) included several known genes involved in hippocampal function, such as the neuron-specific RNA-binding protein Elavl4 and the immediate early gene Egr1 (*Akamatsu et al., 2005*; *Jones et al., 2001*). Consistent with the transient increase in dendritic spines, no transcript identified at 3 days post- exercise remained elevated at 7 days (*Figure 3—figure supplement 2*).

We focused on Mtss1-like (metastasis-suppressor 1-like, Mtss1L), a protein that has been little studied in the adult nervous system (*Saarikangas et al., 2008*). *Mtss1L*, (recently renamed to *Mtss2*, MTSSI-BAR domain containing 2) the most enriched transcript in our experiments, was nine-fold elevated compared to non-activated neighboring cells, as validated in exercise TRAPed cells by RT-PCR (*Figure 3D*, *Figure 3—figure supplement 2*). As a control for the object novelty of the running wheel, we exposed Fos-TRAP mice for 2 hr to an identical wheel that was locked, so that the mice could explore but not run. At 3 days post-exposure, there was a small increase above baseline in the number of Fos-TRAPed cells and *Mtss1L* expression (*Figure 3—figure supplement 3*), but substantially less than exercise-TRAPed mice.

To determine the spatiotemporal pattern of Mtss1L expression, we derived KOMP Mtss1L reporter mice (*Skarnes et al., 2011*) in which the endogenous *Mtss1L* promoter drives bacterial beta-galactosidase (*lacZ*) gene expression. LacZ expression was undetectable in the dentate gyrus of KOMP Mtss1L$^{+/-}$ housed in their homecage, suggesting no expression of Mtss1L under baseline conditions. In contrast, exercise-induced LacZ expression peaked at 3 days post-exercise (*Figure 4*), confirming the activity-dependence of Mtss1L expression in the dentate gyrus. Mtss1L belongs to the BAR (Bin, Amphiphysin and Rvs, I-BAR) protein family, a class of proteins that function by promoting curvature in membranes. BAR domain proteins such as amphyphysin promote inward curvatures including invaginations that are involved in endocytosis or synaptic vesicles at presynaptic nerve terminals (*Dawson et al., 2006*). In contrast, I-BAR proteins promote outward structures typically seen in membrane protrusions. Thus, we hypothesized that Mtss1L expression-mediated exercise-dependent formation or rearrangement of synapses in postsynaptic dendrites.

As proof of principle, we first examined the localization of endogenous Mtss1L in hippocampal neurons in vitro following exposure to brain-derived neurotrophic factor (BDNF), which is upregulated by exercise and involved in activity-dependent synaptic plasticity (*Vaynman et al., 2004*; *Wrann et al., 2013*). Mtss1L immunoreactivity was detectable only after BDNF treatment (*Figure 5A*), consistent with the activity-dependent expression pattern we observed in vivo.

Mtss1L colocalized with the somatodendritic marker MAP2 along dendrites and in dendritic spines (*Figure 5A*). To determine whether Mtss1L could induce spine-like protrusions, we compared spine density in cultured hippocampal neurons transfected with mCherry or mCherry-Mtss1L. At DIV 14, *Mtss1L* overexpression markedly increased spine-like protrusions (*Figure 5B*) including filopodia and mushroom spine subtypes (*Figure 5—figure supplement 1*). The somatodendritic pattern of overexpressed mCherry-Mtss1L overlapped the endogenous Mtss1L expression (*Figure 5B*, yellow). mCherry-Mtss1L-expressing dendritic spines also were labeled with co-transfected PSD-95-FingR-GFP, (Fibronectin Intrabody generated with mRNA display) that binds to endogenous PSD-95 (*Gross et al., 2013*). PSD-95 intrabody labeling of excitatory synapses in living neurons indicated that the protrusions induced by Mtss1L also contain postsynaptic proteins (*Figure 5C*). In vivo, overexpression of mCherry- Mtss1lL by DNA electroporation at P0 increased spine density of mature

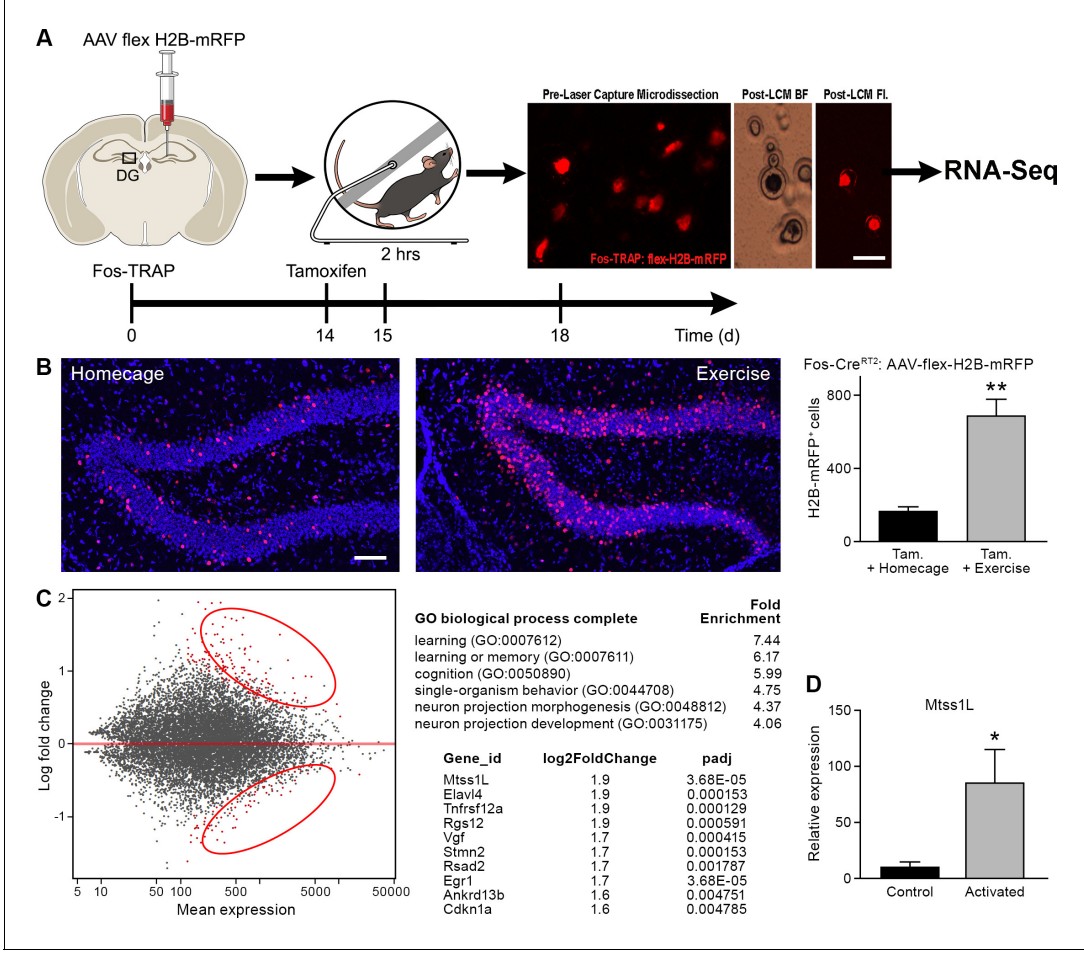

**Figure 3.** Transcriptome analysis of laser captured dentate granule cells activated by a single bout of voluntary exercise. (**A**) A virus expressing a CAG promoter followed by a loxP-flanked ('floxed') stop cassette-controlled Histone2B Monomeric Red Fluorescent Protein (flex H2B-mRFP), was injected stereotaxically into the dentate gyrus of Fos-TRAP mice. The nuclear tag allowed preservation of the fluorophore during laser capture microdissection. Two weeks post-viral injections, mice were injected with tamoxifen, followed 24 hr later by 2 hr of voluntary exercise. Mice were sacrificed 3 and 7 days following exercise. mRFP[+] (exercise TRAPed) and non-activated granule cells were subsequently excised from unfixed intact tissue cryosections using laser capture microdissection, and pooled in batches of 100–150 cells per mouse. cDNA libraries were prepared and samples were processed for RNASeq library construction. Scale bar: 20 μm. (**B**) Voluntary exercise increased activated mRFP[+] cells compared to littermate controls in homecage. Scale bars: 100 μm. Fos TRAPed:H2B-mRFP[+] cells/50 μm section was 164 ± 26 for homecage (n = 3) and 686 ± 92 for exercise (n = 4, unpaired t-test, p=0.007). (**C**) Differential expression of genes between RFP[+] and RFP[-] cell ensembles from four mice are displayed in a MA-plot (M value vs A value plot, which are Log2fold vs normalized mean expression in DEseq2), with and significantly changed (FDR < 0.1, fold enrichment 2.) Upregulated and downregulated transcripts are shown as red dots (upper and lower circles, respectively). For analysis at 3 days post-exercise in exercise-TRAPed cells, the top enriched ontological clusters are listed as well as the top 10 upregulated genes. (**D**) Real-time qPCR confirmation of the Mtss1L expression in in exercise-TRAPed cells at 3 days (see also *Figure 3—figure supplement 2*).

DOI: https://doi.org/10.7554/eLife.45920.007

The following figure supplements are available for figure 3:

**Figure supplement 1.** Filtering, and dispersion estimation of sequencing data used for transcriptomics.
DOI: https://doi.org/10.7554/eLife.45920.008
**Figure supplement 2.** Real-time qPCR confirmation of upregulated genes in exercise-TRAPed cells.
DOI: https://doi.org/10.7554/eLife.45920.009
**Figure supplement 3.** Effect of exposure to a fixed wheel on neuronal activity and Mtss1L expression.
DOI: https://doi.org/10.7554/eLife.45920.010

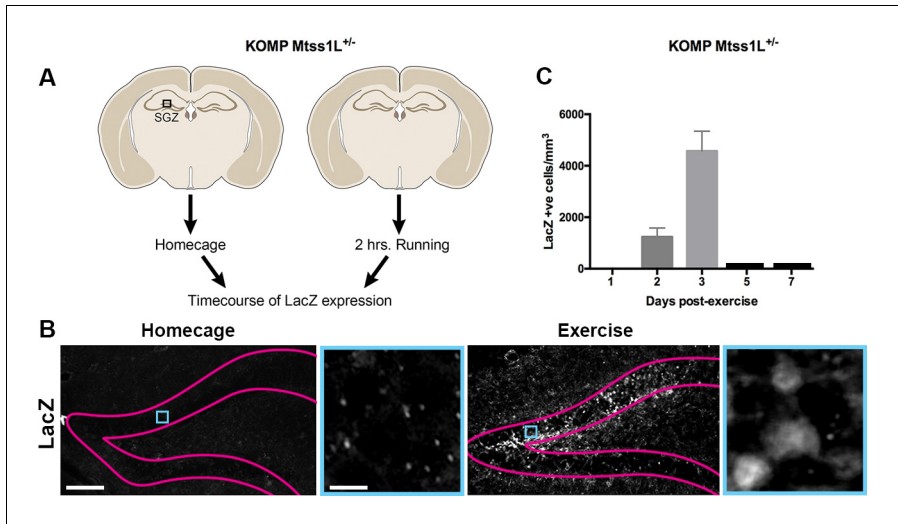

**Figure 4.** Timecourse of LacZ expression in Mtss1L reporter mice. (**A**) Schematic representation of the experimental paradigm used for the Mtss1L reporter mice. KOMP Mtss1l $^{+/-}$ were either housed in their homecage or exposed to 2 hr of voluntary exercise. Brain sections were processed for LacZ immunohistochemistry at several time points. (**B**) Pink lines outline the granular cell layer of the dentate gyrus based on DAPI staining and blue square identifies the area magnified at right. LacZ was detected only in dentate granule cells of mice at 3 days post-exercise (bottom, right), whereas no expression was observed in homecage littermates (bottom, left). Scale bars: 250 μm, 12 μm. (**C**) LacZ expression in granule cells peaked at 3 days post-exercise and was not detectable at 5 or 7 days post-exercise (LacZ+ cells/mm$^3$ 2 days post-exercise: 1235 ± 345, n = 3; 3 days post-exercise: 4573 ± 767,n = 3, one-way ANOVA p<0.0001).

DOI: https://doi.org/10.7554/eLife.45920.011

dentate granule cells at P21 (*Figure 5D*). These results indicate that Mtss1L is expressed in dendrites and can induce spine-like protrusions, consistent with a putative role in synaptic plasticity.

To determine whether Mtss1L was responsible for the increase in synapses in exercise-TRAPed cells, we used a lentivirus expressing shRNAs targeting *Mtss1L*. Validating the efficacy of the shRNAs, shMtss1L reduced mCherry-Mtss1L mRNA and protein expression in HEK293T cells and in primary hippocampal neurons treated with BDNF (*Figure 6—figure supplement 1*). To knockdown *Mtss1L* in vivo (*Figure 6A*), a combination of two lentiviral shMtss1L-GFP constructs or a shscramble-GFP were injected into the dentate gyrus of Fos-TRAP mice. Three weeks post-viral injection, we assessed spine density and excitatory synaptic function in TdT$^+$ (exercise-TRAPed) granule cells that also co-expressed GFP as a marker for shRNA expression. At 3 days post-exercise, knockdown of *Mtss1L* occluded the exercise-induced increase in dendritic spines in the OML (*Figure 6B* middle panel top, and right panel). The shscramble-GFP, injected into the contralateral dentate gyrus of the same mice, had no effect (*Figure 6B* right panel). Viral knockdown of *Mtss1L* in exercise-TRAPed cells also prevented the increase in evoked EPSCs in simultaneous paired recordings from TdT$^-$ and TdT$^+$/GFP$^+$ granule cells (*Figure 6C–D*). The amplitude of EPSCs following viral knockdown in exercise-TRAPed cells was the same as neighboring cells that were not activated by exercise (*Figure 6D*, right panel). Intrinsic membrane properties were not affected by Mtss1L knockdown (*Supplementary file 1*). *Figure 6E* includes the results from *Figure 2* ('exercise-TRAPed OML' (red) and 'exercise-TRAPED MML' (black), compared to the effects of Mtss1L shRNA (yellow), demonstrating that knockdown of Mtss1L was necessary to explain the exercise-induced increase in excitatory synapses. Importantly, evoked EPSCs in granule cells that only expressed GFP +shRNA were unaffected, demonstrating that the effect of Mtss1L was dependent on activity-dependent expression (*Figure 6—figure supplement 2*).

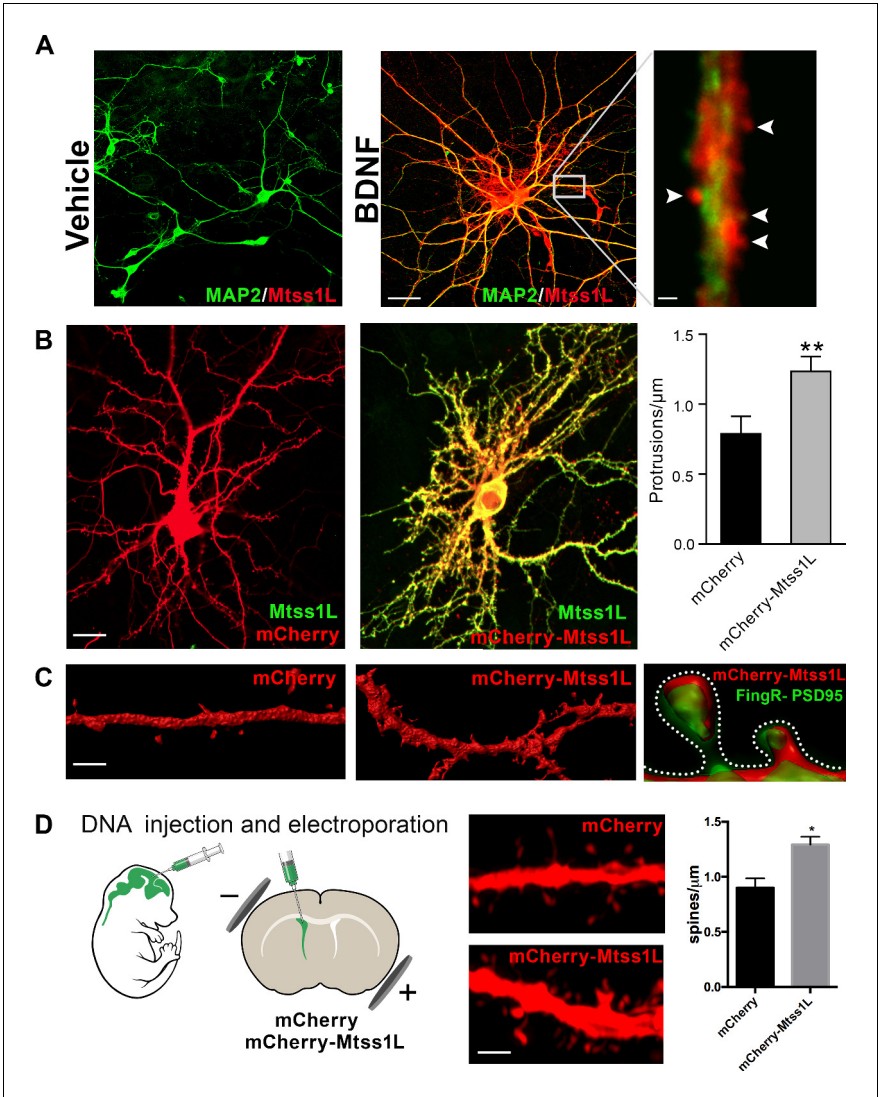

**Figure 5.** Induced Mtss1L expression in vitro and in vivo. (A) Primary hippocampal neurons were cultured with or without BDNF from DIV 7–14 to examine activity-dependence of Mtss1L expression. Representative images of hippocampal neurons co-stained with anti-Mtss1L (red) and the somatodendritic marker anti-MAP2 (green) showed robust Mtss1L immunoreactivity in BDNF-treated cells (middle), but not in the vehicle-treated controls (left). Scale bar: 20 µm. High magnification image of a single dendrite in a BDNF-treated neuron showed localization of Mtss1L (red) in dendritic shaft and spines (right, arrowheads, scale bar: 1.5 µm). (B) Representative images of cultured hippocampal neurons transfected with mCherry or Mtss1L-mCherry and stained with anti-Mtss1L (green). Scale bar: 12 µm. Ectopic expression of Mtss1L markedly increased the number of dendritic protrusions (mCherry: 0.9 ± 0.09, Mtss1L-mCherry: 1.3 ± 0.07, unpaired t-test, p=0.004, n = 3). (C) Higher magnification images of dendritic segments from hippocampal neurons transfected with mCherry (left) or Mtss1L-mCherry (middle) shows the increased number of protrusions. Scale bar: 3 µm. Merged image at right of PSD-95. FingR-GFP and Mtss1L-mCherry in co-transfected neurons demonstrates that the protrusions contained postsynaptic proteins. Scale bar: 0.5 µm. (D) DNA solution was injected into the lateral ventricle of P0 pups followed by gene delivery by electroporation. Representative dendritic segments of dentate granule cells expressing control plasmid mCherry (top) or mCherry-Mtss1L (bottom) 21 days post-electroporation. Scale bar: 3 µm. Mtss1L-mCherry expressing cells show increased dendritic protrusions in vivo (mCherry, 0.9 ± 0.03, n = 3, Mtss1L-mCherry, 1.3 ± 0.07, n = 3, unpaired t-test, p=0.004).
DOI: https://doi.org/10.7554/eLife.45920.012

The following figure supplement is available for figure 5:

**Figure supplement 1.** Effect of Mtss1L overexpression on dendritic spine subtypes of primary hippocampal neurons in vitro.

*Figure 5 continued on next page*

*Figure 5 continued*

DOI: https://doi.org/10.7554/eLife.45920.013

## Discussion

Our experiments were designed to test the cellular and molecular response to acute exercise with an emphasis on time periods during which synapses might form or reorganize. This approach differs from studies of sustained or chronic exercise that likely involve systemic as well as neural mechanisms (*Vivar et al., 2013*; *Voss et al., 2013*; *van Praag and Christie, 2015*). Such studies have shown effects of exercise on many aspects of hippocampal function, including neurogenesis, synaptic plasticity, structural changes at synapses and dendritic spines as well as behavioral effects on learning and memory (*Mandolesi et al., 2018*; *Saraulli et al., 2017*). Our goal was to examine the transcriptional response in the days following acute exercise, which led to the identification of a novel effector of structural plasticity, Mtss1L.

Although the Fos-TRAP method is limited by the time required for reporter expression (ca. 1 day post-exercise), the immediate early gene Egr1 was upregulated (*Figure 3C*), indicating that our protocol detected activity-dependent gene expression in mature granule cells. Although introduction of the running wheel for 2 hr itself could act as a novel object, there was a small increase in Fos-TRAPed cells when the wheel was fixed to prevent running. These results indicate that exercise is a particularly strong stimulus for activation of granule cells, but as expected other stimuli can also activate dentate granule cells (*Ryan et al., 2015*). Adult-generated granule cells in the subgranular zone were not activated by 2 hr of exercise (data not shown). Although it is well known that chronic exercise increases adult neurogenesis (*van Praag, 2008*), progenitors and newborn neurons do not receive direct excitatory perforant path input for several weeks (*Overstreet et al., 2004*), perhaps explaining their lack of activation in our experiments.

Our results provide the first evidence for activity-dependent expression of an I-BAR protein as well as a role in experience-dependent remodeling of synapses. However, I-BAR proteins including MIM (Mtss1L) and IRSp53 can affect constitutive dendritic spine formation (*Saarikangas et al., 2015*; *Choi et al., 2005*). For example, MIM/Mtss1, is involved in cerebellar synapse formation via PIP2-dependent membrane curvature and subsequent Arp2/3-mediated action polymerization (*Saarikangas et al., 2015*). Expression of Mtss1L in neurons has gone largely undetected, likely a result of the previously unrecognized activity-dependence. However, Mtss1L has been detected in radial glial cells during development and implicated in glial membrane protrusions and end-feet (*Saarikangas et al., 2008*). I-BAR protein function at synapses is thought to be mediated by several interaction domains including the N-terminal I-BAR domain and a C-terminal actin-monomer-binding WH2 domain (*Zhao et al., 2011*), and may depend on slice variants. For example, the effects of MIM in cerebellar neurons is splice-variant specific (*Sistig et al., 2017*). Mtss1L also has two known splice variants, both of which were targeted in our overexpression and knockdown experiments. It will be interesting to investigate how protein-protein interactions and splice variants of Mtss1L contribute to their role in synaptic remodeling.

At a circuit level, the laminar-specific increase in synaptic function suggests that acute exercise had a network-specific effect. Entorhinal inputs to the outer molecular layer, the distal dendrites of mature granule cells, have been previously associated with contextual information and particularly the 'how' and 'when' aspects of encoding memories (*Tsao et al., 2018*). How might these results relate to the beneficial effects of exercise? One possibility consistent with our data is that exercise acts as a preconditioning signal that primes exercise-activated neurons for contextual information incoming during the several days following exercise. This represents a broader time window than is usually associated with short-term plasticity. For example, human studies give support for the idea that exercise within 4 hr of a learning task improved memory performance (*van Dongen et al., 2016*). It will be interesting to examine if exercise enhances the pattern of granule cell responses to spatial or context-specific tasks. Our identification of Mtss1L as an activity-dependent I-BAR domain protein makes it ideally suited to act as an early mediator of structural plasticity following neural activity.

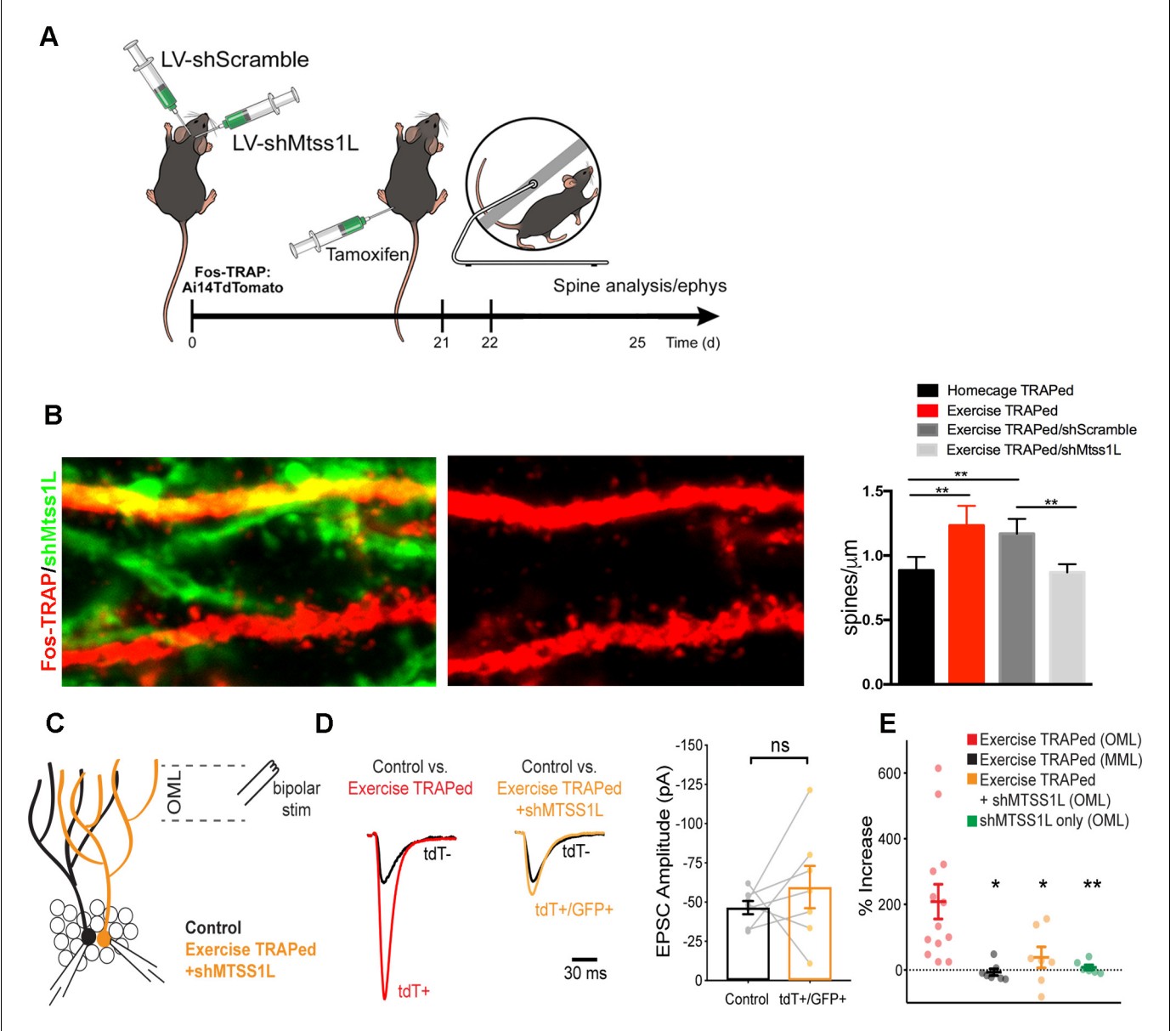

**Figure 6.** Mtss1L knockdown in exercise-TRAPed cells blocked the exercise-induced increase in dendritic spines and synaptic activity at 3 days post-exercise. (A) Effects of Mtss1L knockdown on the spine density of exercise-TRAPed cells were assessed by stereotaxically injecting a control shScramble-GFP lentivirus into the left hemisphere and a shMtss1L-GFP lentivirus in the right hemisphere of Fos-TRAP mice. After 21 days, dentate granule cells were exercise-TRAPed and analyzed 3 days later as shown in the schematic. (B) The left panel shows representative OML dendritic segments of exercise-TRAPed cells (red), exercise-TRAPed cells co-expressing shRNA (orange) next to several dendrites expressing only shRNA (green). The middle panel shows the dendritic field isolated in the red channel to allow comparison of dendritic spines in exercise-TRAPed cells at 3 days post exercise with (top, middle panel) and without (bottom, middle panel) co-expression of the Mtss1L shRNA. Mtss1L shRNA, but not shScramble, blocked the exercise–induced increase in dendritic spines in exercise-TRAPed cells. Summary graph at right shows dendritic spine density in the dentate OML for each condition (Homecage TRAPed: 0.8 ± 0.1, n = 4, Exercise TRAPed: 1.2 ± 0.2, n = 5, Exercise TRAPed/shScramble: 1.17 ± 0.1, n = 4, Exercise-TRAPed/shMtss1L: 0.87 ± 0.06, n = 5, one-way ANOVA, multiple comparisons Dunnett's test, p<0.01). (C) Schematic of simultaneous whole-cell recordings from control (black) and exercise-TRAPed cells co-expressing shRNA-Mtss1L (orange) to assess functional synaptic activity. Lateral perforant path axons were stimulated in the OML. (D) Superimposed EPSCs from a representative control cell (black) and a simultaneously recorded exercise-TRAPed cell (red) showed a large increase in amplitude in the exercise-TRAPed cells, as quantified in **Figure 1E**. In contrast, superimposed EPSCs from a control cell (black, 46.5 ± 4.2 pA) and an exercise-TRAPed cell co-expressing shMtss1L (orange, −59.6 ± 13.5 pA) showed no increase in amplitude (p=0.42, paired t-test, seven-cell pairs from five mice). Traces are normalized and scaled relative to control cells (black) for presentation. (E) Summary plot across experimental conditions. The exercise-TRAPed increase in EPSC amplitude in the OML was blocked by Mtss1L shRNA expression in exercise-TRAPed cells. (Percent increase in EPSC amplitudes: exercise-TRAPed - OML stimulation, 208.2 ± 52.8%, n = 13 cells, eight mice; exercise-

*Figure 6 continued on next page*

*Figure 6 continued*

TRAPed - MML stimulation, - 6.7 ± 10.1%, n = 7 cells, four mice; exercise TRAPed +shMtss1L - OML stimulation, 38.4 ± 32.1%, n = 7 cells, five mice; shMtss1L only - OML stimulation, 7.8 ± 8.0%, n = 6 cells, three mice, p=0.002, one-way ANOVA; exercise-TRAPed shMtss1L, p=0.025; shMtss1L only, p=0.011; exercise-TRAPed MML, p=0.004; Dunnett's test).

DOI: https://doi.org/10.7554/eLife.45920.014

The following figure supplements are available for figure 6:

**Figure supplement 1.** Mtss1L shRNA knockdown efficiency.

DOI: https://doi.org/10.7554/eLife.45920.015

**Figure supplement 2.** shRNA-Mtss1L did not alter evoked EPSCs in granule cells that were not exercise- TRAPed.

DOI: https://doi.org/10.7554/eLife.45920.016

## Materials and methods

### Mice

All procedures were performed according to the National Institutes of Health Guidelines for the Care and Use of Laboratory Animals and were in compliance with approved IACUC protocols at Oregon Health and Science University. Both female and male mice from a C57BL6/J background were used for experiments, aged 6–8 weeks at the time of surgery. TRAP mice were heterozygous for the $cFos^{creERT2}$ allele (*Guenthner et al., 2013*); some were also heterozygous for the *B6;129S6-Gt(ROSA)26Sortm14(CAG-tdTomato)/Hze/J(Ai14*;JAX#007908) allele for experiments involving tdTomato labeling. Cryopreserved Mtss1l$^{tm1a(KOMP)Wtsi}$ sperm was obtained from the Knockout Mouse Project (KOMP) Repository (The Knockout Mouse Project, Mouse Biology Program, University of California, Davis, CA) under the identifier CSD79567. The Mtss1l$^{tm1a(KOMP)Wtsi}$ mouse line was re-derived by in vitro fertilization at the OHSU transgenic core. Genotype was confirmed by PCR. *Mtss1L* (recently renamed to *Mtss2*, MTSS I-BAR domain containing 2) deletion in Mtss1L$^{-/-}$ KO KOMP mice was confirmed by RT-qPCR and immunohistochemistry in the cerebellum. Cerebellar Purkinje cells, perhaps because of their high basal firing rate, are the only cells in the brain with detectable basal Mtss1L expression. Mtss1L mRNA and protein levels were undetectable in the cerebellum of Mtss1L$^{-/-}$ KO KOMP mice. The KOMP allele contains a Neo cassette driven by a mouse phosphoglycerate kinase promoter (PGK), which is bidirectional and may interefere with gene expression. Thus, for the generation of the Mtss1L$^{+/-}$ reporter mice, the Neo cassette was removed in a tissue-specific manner by stereotaxically injecting an AAV nuGFP-Cre PGK into the dentate gyrus of Mtss1L$^{-/-}$ KO KOMP mice 3 weeks prior to experiments. All mice were housed in plastic cages with disposable bedding on a standard light cycle with food and water available ad libitum.

### Stereotaxic injections

Mice were anesthetized using an isoflurane delivery system (Veterinary Anesthesia Systems Co.) by spontaneous respiration, and placed in a Kopf stereotaxic frame. Skin was cleaned with antiseptic and topical lidocaine was applied before an incision was made. The dentate gyrus was targeted by placing Burr holes at x:±1.1 mm, y: −1.9 mm from bregma. Using a 10 µl Hamilton syringe with a 30 ga needle and the Quintessential Stereotaxic Injector (Stoelting), 2 µl mixed viral stock (1 µl of each virus) was delivered at 0.25 µl/min at z-depths of 2.5 and 2.3 mm. The syringe was left in place for 1 min after each injection before slow withdrawal. The skin above the injection site was closed using veterinary glue. Animals received post-operative analgesia with topical lidocaine and flavored acetaminophen in the drinking water.

### TRAP induction

Tamoxifen was dissolved at 20 mg/ml in corn oil by sonication at 37°C for 5 min. The dissolved tamoxifen was then stored in aliquots at –20°C for up to several weeks or used immediately. The dissolved solution was injected intraperitoneally at 150 mg/kg. For exercise-TRAP, tamoxifen was administered 23 hr before exposure to the running wheel. Experiments were begun 48 hr after tamoxifen administration to allow for reporter expression.

## Voluntary exercise and fixed wheel protocol

Both homecage and exercise groups were housed together until 7 days before the experiment, then they were singled housed in an oversized (rat) sedentary cages (43 × 21.5 cm$^2$) to allow acclimation in the novel environment before tamoxifen administration. At 23 hr post-tamoxifen injection, a running wheel (free or locked) was introduced in the cage of animals in the exercise group and mice had free access to running or exploring for 2 hr at the beginning of the dark period, after which the running wheel was removed from the cage. Total distance (km) was measured using an odometer. All mouse groups were handled for 5 days before tamoxifen administration.

## Immunohistochemistry

Mice were terminally anesthetized, transcardially perfused with saline and 20 ml of 4% paraformaldehyde (PFA), and brains were post-fixed overnight. Coronal sections (100 µm) of the hippocampus were collected and permeabilized in 0.4% Triton in PBS (PBST) for 30 min. Sections were then blocked for 30 min with 5% horse serum in PBST and incubated overnight (4°C) with primary antibody in 5% horse serum/PBST. After extensive washing, sections were incubated with the appropriate secondary antibody conjugated with Alexa 488, 568 or 647 (Molecular Probes), for 2hrs at room temperature. They were then washed in PBST (2 × 10 min) and mounted with Dapi Fluoromount-G (SouthernBiotech). The primary antibodies used were: anti-c-fos (1:500, Santa-Cruz), anti-tdTomato (1:500, Clontech), anti-MAP2 (1:500, Sigma), anti-ABBA/Mtss1L (1:500, Millipore), anti-VGLUT2 (1:500, Synaptic Systems) and anti-beta-galactosidase (1:20, Developmental Studies Hybridoma Bank). All antibodies have been well characterized in prior studies in our laboratory, and staining was not observed when the primary antibody was omitted.

## Imaging and morphological analysis

For imaging we used an LSM 7 MP laser scanning microscope (Carl Zeiss MicroImaging; Thornwood, NY). All cell and dendritic spine counting and dendritic arbor tracing were done manually with ImageJ and Imaris (National Institutes of Health; Bethesda, MD). For quantification of immunopositive cells, six hippocampal slices at set intervals from dorsal to ventral were stained per animal; 3–6 animals were analyzed per group. A 49 µm z-stack (consisting of seven optical sections of 7 µm thickness) was obtained from every slice. Positive cells were counted per field from every z-stack, averaged per mouse and the results were pooled to generate group mean values. For analysis of spine density of dentate granule cells, sections were imaged with a 63x objective with 2.5x optical zoom. The span of z-stack was tailored to the thickness of a single segment of dendrite with 0.1 µm distance between planes. Dendritic spines were imaged and analyzed in the middle (MML) and outer molecular (OML) layers of the dentate gyrus. The MML and OML were distinguished based on the pattern of VGluT2 immunofluorescence, which begins at the border between the inner molecular layer and middle molecular layer. MML dendritic segments were therefore imaged at the beginning of the VGluT2 staining nearest the granule cell body layer, whereas OML dendritic segments were imaged at the distal tip of the molecular layer. The same microscope settings (laser intensity and gain) were used for each experimental group analyzed. Slides were coded and imaged by an investigator blinded to experimental condition.

## Laser Capture Microdissection (LCM)

To obtain nuclei of exercise-TRAPed cells, fresh frozen coronal brain sections (12 µm) were collected by cryostat sectioning on polyethylene napthalate membrane slides. Sections were fixed for 1 min in 75% ethanol, followed by a final wash in 100% ethanol for 1 min. Pools of 150 RFP$^+$ and RFP$^-$ individual cells were laser captured from the granular cell layer of the dentate gyrus of each mouse using a Leica Microsystem LMD system under a HC PL FL L 63/0.60 XT objective and collected in lysis buffer (50 µl RLT buffer, Qiagen). Total RNA from LCM samples was isolated separately from each animal.

## RNA extraction and RNAseq library preparation

RNA was extracted from laser-captured pooled nuclei using RNeasy mini-elute column with a modified protocol. Briefly, cells were picked from laser capture microscopy onto caps of 0.5 ml Eppendorf tubes, 50 µl RLT buffer was added immediately, and placed on ice. After spinning down the samples, 300 µl RLT was added to each sample, mixed well and vortexed for 30 s. After a brief spin, 2 ng

tRNA was added as carrier and 625 µl 100% ETOH was added into each sample and mixed well. After a quick spin, samples were loaded onto RNeasy mini-elute columns, washed, dried and eluted in 10 µl RNAse-free water. 8 µl RNA was used for reverse transcription using Superscribe reverse transcriptase 2 ul, DTT 0.1M 0.5 ul, 5x RT buffer 4 µl at 42°C for 1.5 hr then heat inactivated at 75°C for 20 min (self-developed protocol for RNA extraction, superscribe reverse transcriptase was from Clontech, protocol adopted from manufacturer's protocol for smarter single-cell RNAseq v4). TSO (AAGCAGTGGTATCAACGCAGAGTACrG +GrG) was ordered from Exicon. After RNA, PCR was carried out using Seqamp DNA polymerase and amplification PCR primer (AAGCAGTGGTA TCAACGCAGAGTAC). Half of amplified cDNA was used for qPCR and the other half made into sequencing libraries using Nextera XT DNA library prep kit (Illumina) Quality of cDNA libraries were checked on Bioanalyzer using DNA high sensitivity Chip. 2 nM library dilutions were mixed and denatured, 1.8 pmol mixed library was sequenced on Nextseq500 (Illumina). Libraries were sequenced to a depth of 14–30 million reads. Alignment rate of total reads was above 90% across all samples. Gene level coverage at 1x was ~10000 genes and even among all samples with no 3' or 5' end bias.

## RNAseq analysis

After sequencing, sequencing quality was checked by fastqc, raw reads were aligned with STAR 2.5b on Illumina Basespace. Raw reads were then plotted as a violin plot using the R ggplot2 package. Genes with reads lower than 64 in all samples were filtered away and dataframe was replotted to check data distribution. Individual samples that differed significantly in their count distribution due to technical issues were discarded from further analysis. Data dispersion and differential expression analysis was done on filtered reads using DEseq2 unpaired analysis. FDR and <0.15 fold enrichment and > 1.5 fold were considered as enriched genes and were plotted in MA-plot using R script. Genes enriched in TRAPed cells at 3 days post exercise were further analyzed by PantherGO for GO terms enrichment (release 20160321, GO Ontology database released 2016-05-20); protein-protein reactions analyzed by Cytoscape application Biogenet.

## Real-time qPCR

To quantify gene expression levels, amplified cDNA was mixed with the iQ5 SYBR Green PCR master mix (Biorad) and 5 pmol of both forward and reverse primers were used. For qPCR of our miRNA-based shRNA, miRNA was converted to cDNA using the QuantiMiR reverse transcription kit (System Biosciences, Mountain View, CA). Briefly, RNA was polyadenylated with ATP by poly(A) polymerase at 37°C for 1 hr and reverse transcribed using 0.5 µg of poly(T) adapter primer. Each miRNA was detected by the mature DNA sequence as the forward primer and a 3' universal reverse primer provided in the QuantiMir RT kit. Human small nuclear U6 RNA was amplified as an internal control. qPCR was performed using Power SYBR Green PCR Master Mix (Applied Biosystems). All qPCR performed using SYBR Green was conducted at 50°C for 2 min, 95°C for 10 min, and then 40 cycles of 95°C for 15 s and 60°C for 1 min. The specificity of the reaction was verified by melt curve analysis. Relative expression was calculated using mouse Ubc or 18S as internal controls as specified in the Figure legends. A table of primers is included as *Supplementary file 4*.

## Primary neuronal hippocampal cultures and in vitro transfections

Hippocampi from E18-19 embryos were isolated as described previously (*Seibenhener and Wooten, 2012*) and placed on ice. The tissue was chopped into 1 mm cubes with fine scissors and transferred to a 15-ml tube using a Pasteur pipette. This tube was placed in a water bath at 37°C with the papain solution, separately, and both were allowed to equilibrate at this temperature for 5 min. The tissue was then carefully transferred into the papain solution and placed on an orbital shaker at 100 rpm at 37°C for 15 min. The tissue was then gently transferred from the papain solution into a 15-ml tube containing 2 ml HBSS. This step was repeated 2X to remove any residual papain. The tissue was then triturated using fire polished Pasteur pipettes. Following trituration and centrifugation, the single-cell suspension was seeded on poly- D-Lysine coated coverslips in 24-well plates at $10^4$ cells/well in Neurobasal Plating Media with B27 Supplement [1 ml/50 ml], 0.5 mM glutamine solution, penicillin(10,000 units/ml)/streptomycin (10,000 µg/ml) [250 µl/50 ml], 1 mM HEPES (MW238.3 g/mol), 10%

Heat-Inactivated horse serum. HI-horse serum was removed by half- changes of the plating media beginning at 24 hr.

For transfection of cultured neurons, DNA-Lipofectamine 2000 complexes were prepared as follows: 1 µg of DNA was diluted in 50 µl of Neurobasal medium and 2 µl of Lipofectamine 2000 was diluted in another 50 µl of Neurobasal medium. Five min after dilution of Lipofectamine 2000, the diluted DNA was combined with the diluted lipid. The solutions were gently mixed and then left for 20 min at room temperature to allow the DNA–Lipofectamine 2000 lipoplexes to form. 100 µl of transfection complex was added to each well containing cells and medium, and the cells were incubated at 37°C in a humidified incubator with 5% $CO_2$ for 1 hr after which transfection media was changed. Transfection efficiency was assayed 48 hr post-transfection by fluorescence microscopy.

## In vivo electroporation into postnatal granule cells of the dentate gyrus

Postnatal day 0 (P0) electroporation of the dentate gyrus was performed according to a previously published method (*Ito et al., 2014*). In brief, mice were anesthetized by hypothermia, and then a sharp glass electrode with a beveled tip containing plasmid (1 µg/µl in TE mixed with 0.05% Fast Green) was inserted through skin and skull. One µl of DNA solution was injected into the lateral ventricle (LV). Correct injection was confirmed by trans-cranial visualization of Fast Green in the LV in the dissecting microscope. Successfully injected pups were immediately electroporated with tweezer electrodes (5 mm platinum) positioned onto the brains as described by *Ito et al. (2014)*. Five pulses of 100 V and of 50 ms duration were given at 950 ms intervals. Electroporated animals were placed in a recovery chamber at 37°C for several minutes and then returned to their mother.

## Plasmid construction

PLVX-cherry-C1 plasmid was from Clontech and Mtss1L coding region was cloned into the C-terminus of mCherry between BsrG1 and SmaI sites to generate fusion construct of PLVX-cherryC1-Mtss1L (Mtss1L coding region sequence Accession number B), FUCW-T2A-Mtss1L was constructed using NEB HiFi DNA builder assembly kit. shABBA constructs were either obtained from Dharmacon or designed by Biosettia and subcloned into PLV-Ubc-GFP-mU6-shRNA using NEB HIFI DNA assembly mix. The folllowing shRNA sequences were used: Mtss1L1:AAAAGGCCGTTTCTGCACCTTTA TTGGATCCAATAAAGGTGCAGAAACGGCC, Mtss1L2 (miRNA-based shRNA): TCTCTACTAGGCTG TGCCT, Scramble:AAAAGCTACACTATCGAGCAATTTTGGATCCAAAATTGCTCGATAGTGTAGC.

## Slice physiology

Acute brain slices were prepared as previously described (*Hendricks et al., 2017*). In summary, 8-week-old male and female mice were anesthetized with 4% isoflurane and injections of 1.2% avertin (Sigma-Aldrich). Mice were perfused with 10 mL of ice-cold cold N-methyl-D-glucamine (NMDG)-based cutting solution, containing the following (in mM): 93 NMDG, 30 NaHCO$_3$, 24 glucose, 20 HEPES, 5 Na-ascorbate, 5 N-acetyl cysteine, 3 Na-pyruvate, 2.5 KCl, two thiourea, 1.2 NaH$_2$PO$_4$, 10 MgSO$_4$, and 0.5 CaCl$_2$. Transverse 300-µm-thick hippocampal sections were cut in ice-cold NMDG solution on a Leica VT1200S vibratome. Slices recovered in warm (34°C) NMDG solution for 15 min. Slices were transferred to room temperature in standard ACSF and allowed to recover for at least 1 hr prior to recording. Dentate granule cell whole-cell recordings were made with 3–5 MΩ borosilicate glass pipets filled with a Cs$^+$-based internal solution containing (in mM): 113 Cs-gluconate, 17.5 CsCl, 10 HEPES, 10 EGTA, 8 NaCl, 2 Mg-ATP, 0.3 Na-GTP, 0.05 Alexa Fluor 488, pH adjusted to 7.3 with CsOH. Osmolarity was adjusted to 295 mOsm and QX-314-Cl (5 mM; Tocris Bioscience) was included to block unclamped action potentials. The junction potential was 8 mV and left uncorrected. Granule cells were visually identified on an Olympus BX-51WI microscope using infrared differential interference contrast imaging. To ensure relatively equal stimulation intensity between cells during paired simultaneous tdT +and tdT- recordings, neighboring tdT +and tdT- (control) cells were selected. Cells were only considered 'neighbors' if they were within roughly ±2 cell bodies distance apart in the x and y planes, and roughly ±1 cell body distance in the z plane. Granule cells were filled with AlexaFluor 488 to ensure dendritic overlap. Bi-polar stimulating electrodes (FHC) were placed in distally in the OML and medially in the MML, to ensure OML and MML-specific stimulation of lateral and medial perforant path, respectively (*Woods et al., 2018*). Electrical stimulation was delivered with a constant current stimulator (Digitimer, Inc); intensity was tittered to produce

50–100 pA EPSCs in tdT- cells. Proximity to the stimulating electrode and recording configuration was randomly assigned. Cells were voltage-clamped to −70 mV and signals were amplified using two AxoClamp 200B amplifiers (Molecular Devices), Bessel filtered at 5 kHz, and captured at 10 kHz with an analog-to-digital converter (National Instruments). Data were collected and analyzed in Igor Pro eight using the Igor NIDAQ Tools MX script package (Wavemetrics). Averages were the mean of 15 consecutive sweeps.

### Cell lines and culture

HEK293T cells were purchased from CH3 Biosystems (#920001) and maintained in DMEM media supplemented with 10% fetal bovine serum, non-essential amino acid and penicillin-streptomycin using standard procedures.

The identity of the HEK 293 T cell line was provided and authenticated by CH3 Biosystems (#920001), where the line was purchased from. Mycoplasma screening was negative.

### Statistical analysis

Sample sizes were based on pilot experiments with an effect size of 20% and a power of 0.8. Littermates were randomly assigned to control or exercise groups. Criteria were established in advance based on pilot studies for issues including data inclusion, outliers, and selection of endpoints. Criteria for excluding animals from analysis are listed in the methods. Mean ± SD was used to report statistics for all apart from electrophysiology experiments where Mean ± SE was used. The choice of statistical test, test for normality, definition of N, and multiple hypothesis correction where appropriate are described in the figure legends. Unless otherwise stated, all statistical tests were two-sided. Significance was defined as $p < 0.05$. All statistical analyses were performed in Prism or Igor Pro 8.

## Acknowledgements

We thank the following for assistance with our experiments: Stefanie Kaech Petrie, Aurelie Snyder, Crystal Shaw and the Advanced Light Microscopy Core (P30 NS061800); Shannon McWeeney and Sophia Jeng in the OHSU Bioinformatics core; Laura Villasana and Jocelyn Santiago-Perez (behavioral testing); Francesca Cargnin (electroporation); Lev Federov and OHSU transgenic core (generation of KOMP mice). This work was supported by the Ellison Foundation and NIH NS080979 (RHG and GLW), Department of Veterans Affairs Merit Review Award I01-BX002949 (ES); a Department of Defense CDMRP Award W81XWH-18-1-0598 (ES); a National Institutes of Health (NIH) Grant F31-NS098597 (WDH); and a fellowship from Ronni Lacroute (CC). The contents of this manuscript do not represent the views of the U.S. Department of Veterans Affairs or the United States government. RNA seq data generated in the manuscript have been deposited at: https://www.ncbi.nlm.nih.gov/bioproject/PRJNA481775.

## Additional information

### Competing interests

Gary L Westbrook: Senior editor, *eLife*. The other authors declare that no competing interests exist.

### Funding

| Funder | Grant reference number | Author |
|---|---|---|
| National Institutes of Health | NS080979 | Richard H Goodman<br>Gary L Westbrook |
| U.S. Department of Defense | W81XWH-18-1-0598 | Eric Schnell |
| U.S. Department of Veterans Affairs | I01-BX002949 | Eric Schnell |
| Lawrence Ellison Foundation | | Richard H Goodman |
| National Institutes of Health | F31-NS098597 | Wiiliam D Hendricks |

The funders had no role in study design, data collection and interpretation, or the decision to submit the work for publication.

## Author contributions

Christina Chatzi, Conceptualization, Data curation, Formal analysis, Supervision, Funding acquisition, Validation, Investigation, Visualization, Methodology, Writing—original draft, Project administration, Writing—review and editing; Yingyu Zhang, Conceptualization, Data curation, Formal analysis, Supervision, Validation, Investigation, Visualization, Methodology, Writing—review and editing; Wiiliam D Hendricks, Conceptualization, Data curation, Formal analysis, Validation, Investigation, Visualization, Methodology, Writing—review and editing; Yang Chen, Conceptualization, Data curation, Formal analysis, Validation, Investigation, Visualization, Writing—review and editing; Eric Schnell, Conceptualization, Resources, Data curation, Software, Supervision, Funding acquisition, Validation, Investigation, Visualization, Writing—review and editing; Richard H Goodman, Conceptualization, Resources, Data curation, Supervision, Funding acquisition, Validation, Investigation, Visualization, Methodology, Project administration, Writing—review and editing; Gary L Westbrook, Conceptualization, Resources, Data curation, Supervision, Funding acquisition, Validation, Investigation, Visualization, Methodology, Writing—original draft, Project administration, Writing—review and editing

## Author ORCIDs

Christina Chatzi (iD) https://orcid.org/0000-0001-8922-5617
Yingyu Zhang (iD) https://orcid.org/0000-0002-3414-4059
Wiiliam D Hendricks (iD) https://orcid.org/0000-0003-2841-577X
Yang Chen (iD) https://orcid.org/0000-0002-4365-9900
Eric Schnell (iD) https://orcid.org/0000-0002-5623-5015
Gary L Westbrook (iD) https://orcid.org/0000-0002-8108-5223

## Ethics

Animal experimentation: All procedures were performed according to the National Institutes of Health Guidelines for the Care and Use of Laboratory Animals and were in compliance with approved IACUC protocols at Oregon Health & Science University (protocol# TR01-IP00000148). For surgeries, all mice were anesthetized using an isoflurane delivery system (Veterinary Anesthesia Systems Co.) by spontaneous respiration and every effort was made to minimize suffering. All investigators underwent institutional Responsible Conduct & Research training.

## Decision letter and Author response

Decision letter https://doi.org/10.7554/eLife.45920.025
Author response https://doi.org/10.7554/eLife.45920.026

# Additional files

## Supplementary files

• Supplementary file 1. Intrinsic properties of granule cells during paired recordings.
DOI: https://doi.org/10.7554/eLife.45920.017

• Supplementary file 2. Three-days complete gene list DESeq.
DOI: https://doi.org/10.7554/eLife.45920.018

• Supplementary file 3. Seven-days complete gene list DESeq.
DOI: https://doi.org/10.7554/eLife.45920.019

• Supplementary file 4. List of primers used for RT-qPCR analysis.
DOI: https://doi.org/10.7554/eLife.45920.020

• Transparent reporting form
DOI: https://doi.org/10.7554/eLife.45920.021

## Data availability

RNA seq data generated in the manuscript have been deposited at https://www.ncbi.nlm.nih.gov/bioproject/PRJNA481775.

The following dataset was generated:

| Author(s) | Year | Dataset title | Dataset URL | Database and Identifier |
|---|---|---|---|---|
| Zhang Yingyu, Chatzi Christina | 2018 | Brief Running Activated vs Non-activated Neurons RNAseq | https://www.ncbi.nlm.nih.gov/bioproject/PRJNA481775 | NCBI Bioproject, PRJNA481775 |

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
