## [Decision Letter]

[Editors’ note: the authors were asked to provide a plan for revisions before the editors issued a final decision. What follows is the editors’ letter requesting such plan.]

Thank you for sending your article entitled "Exercise-induced enhancement of synaptic function triggered by the inverse BAR protein, Mtss1L" for peer review at *eLife*. Your article is being evaluated by three peer reviewers, and the evaluation is being overseen by a Reviewing Editor and Ronald Calabrese as the Senior Editor.

Given the list of essential revisions, including new experiments, the editors and reviewers invite you to respond with an action plan and timetable for the completion of the additional work. We plan to share your responses with the reviewers and then issue a binding recommendation.

Summary:

The authors present work where Mtss1L is identified as an exercise induced hippocampal gene, then show modulation of Mtss1L levels changes exercise induced spine density. Overall there are few papers on I-BAR genes in the nervous system and even fewer on Mtss1L making this work potentially novel.

The authors address mechanisms contributing to exercise-induced synaptic plasticity in the hippocampus using Fos-TRAP mice to identify exercise-activated dentate granule cells. They show that brief bouts of running result in changes in spine density and synaptic strength, and they perform RNASeq to identify potential molecular mediators. shRNA knock-down of one highly-upregulated gene, Mtss1L, was sufficient to block the exercise-induced change in spines and EPSCs. Thus, they propose Mtss1L is a novel effector of activity-dependent synapse function. Overall the experiments, analysis and interpretations are convincing and the conclusions are interesting.

Essential revisions:

- The authors don't directly show that Mtss1L expression is activity dependent. Their in vitro data suggest it is BDNF dependent, but do not show changes in firing activity cause Mtss1L expression. It is unclear how closely the synaptic changes are associated with exercise rather than then Fos-expressing status of GCs (which could occur under various conditions and in the homecage). Since the authors strongly focus on exercise and subsequent cognitive benefits in the introduction, it is important to confirm that these are exercise-mediated effects.

- Figure 2/Figure 6. All of the functional data involves comparison of TRAPed and non-TRAPed (control) GCs in the exercise condition. To confirm that the effects result from exercise, the author should address functional synaptic differences in TRAPed and control GCs in the non-exercise condition. This would address the possibility that TRAPed cells exhibit increased OML synaptic function in the absence of exercise.

- Figure 4 addresses the expression of Mtss1L in the DG in response to exercise, but the images are not very helpful. Higher magnification images are needed to illustrate the expression pattern. It is also unclear why the authors used the KOMP approach rather than the antibody shown in Figure 5A. Co-labeling of Mtss1L and Fos-TRAP before and after exercise should be obtained to confirm that it is upregulated only in Fos-TRAPed GCs.

- Results, fifth paragraph: Key reagent validation is necessary. The KOMP Mtss1L^tm1a(KOMP)Wtsi^ allele is reporter tagged allele with conditional potential. LacZ activity with this allele is not Cre dependent. LacZ activity can be eliminated with Flp expression. Treating with Cre will excise the Neo cassette and exons 7 and 8 leaving LacZ expressed and unaffected. Of concern is the lack of characterization of the allele and the apparently unnecessary manipulations on their animals (nuGFP-Cre injection). Additional controls need to be included that are required of those making or using an allele for the first time. The authors should more clearly explain the motivation for nuGFP-CRE line of experiments, as it is not clear in the text.

- Additionally, several sequencing and array studies suggest Mtss1L expression in neurons is ambiguous, but high in radial glia (Saarikangas et al., 2008), astrocytes (astrocyternaseq.org), oligodendrocyte progenitors (https://web.stanford.edu/group/barres_lab/brain_rnaseq.html, described in Zhang et al., 2014, J. Neurosci.). Additional experiments to better characterize Mtss1L staining would be incredibly helpful in resolving this ambiguity.

- The Mtss1L paralog Mtss1 have multiple splice variants that are biologically important controlling localization in the cell and other functions. Given the similarity one may expect similar for Mtss1L and clearly documenting the spliceform used is essential. Related, Figure 5C and D – the interpretation of these overexpression experiments will depend strongly on the spliceform of Mtss1L used. Saarikangas et al. (2015) demonstrated the Mtss1 IMD domain alone is sufficient to induce extra spines, which is almost completely identical to Mtss1L. However, Mtss1 and Mtss1L have different overexpression phenotypes, suggesting domains outside the N-terminal IMD region are biologically important. Without explaining what form of Mtss1L is being used it's impossible for readers to determine whether these overexpression phenotypes reflect Mtss1L function, or whether a truncated form is mimicking roles established for Mtss1.

- Figure 6D. The authors show that expressing Mtss1L is sufficient to alter spines in GCs in the absence of exercise, suggesting an important contribution to on-going synaptic plasticity. Thus, it seems counter-intuitive that knockdown of Mtss1L had no effect on EPSCs in the basal condition (Figure 6—figure supplement 2). This should be explained (as it relates to the first point).

[Editors’ note: formal revisions were requested, following approval of the authors’ plan of action.]

Thank you for submitting your article "Exercise-induced enhancement of synaptic function triggered by the inverse BAR protein, Mtss1L" for consideration by *eLife*. Your article has been reviewed by three peer reviewers, and the evaluation has been overseen by a Reviewing Editor and Ronald Calabrese as the Senior Editor. The following individuals involved in review of your submission have agreed to reveal their identity: Paul F Worley (Reviewer #3).

The reviewers have discussed the reviews and your revision plan with one another. The revision plan is generally acceptable, but there is one additional experiment, listed under essential revisions, below, that will bolster the specific claims made in the Abstract and Introduction. The Reviewing Editor has drafted this decision to help you prepare a revised submission.

Essential revisions:

The reviewers were not fully satisfied about the revision response to point 2 relating to the relationship between the authors conclusions about Mtss1L-induced synapse/spine plasticity and exercise.

There is no doubt that Fos+ cells represent an active population of GCs in the DG, sometimes referred to as "engram" cells. The Tonegawa lab has used a similar approach to show that Fos+ GCs have greater evoked EPSCs and spine density compared to Fos- GCs after a 5 min exposure to novel context (see Supplementary Figure 7 of Ryan et al., 2015). The authors should do a similar experiment – repeat the experiment of 2B in mice that have not undergone exercise – to test whether exercise is really required for this main outcome measure. The Ryan et al. data suggests that the synaptic function and spine difference between Fos+ and Fos- cells will exist in the absence of exercise (i.e. exercise is not required). The result of this experiment could differentiate whether Mtss1L is simply an activity-dependent gene downstream of Fos or whether it is specifically related to exercise-induced activity as indicated in the Abstract and Introduction. The outcome of this experiment does not change the mechanistic novelty of the role of Mtss1L in activity-dependent synaptic plasticity, but it could affect the conclusions that the authors make about this mechanism in the context of exercise. At the very least, the authors should discuss their results in the context of what has been reported about synapses/spines of Fos+ and Fos- GCs in the absence of exercise, which is very similar to what they are currently ascribing to an effect of exercise.

It is unclear how closely the synaptic changes are associated with exercise rather than then Fos-expressing status of GCs (which could occur under various conditions and in the homecage). Since the authors strongly focus on exercise and subsequent cognitive benefits in the Introduction, it is important to confirm that these are exercise-mediated effects.

---

## [Author Response]

[Editors’ note: what follows is the authors’ plan to address the revisions.]

Essential revisions:- The authors don't directly show that Mtss1L expression is activity dependent. Their in vitro data suggest it is BDNF dependent, but do not show changes in firing activity cause Mtss1L expression. It is unclear how closely the synaptic changes are associated with exercise rather than then Fos-expressing status of GCs (which could occur under various conditions and in the homecage). Since the authors strongly focus on exercise and subsequent cognitive benefits in the Introduction, it is important to confirm that these are exercise-mediated effects.

We disagree with the interpretation that we do not demonstrate that Mtss1L is activity-dependent. Our experiments were specifically setup to reveal changes occurring over days, not minutes, that is, the typical time frame of activity-dependent gene expression. Whether Mtss1L is induced by short-term activity is not the point; what’s important is that Mtss1L expression occurs over a period of days. So the BDNF response is on point—activity induces expression of BDNF, which then induces expression of Mtss1L. What’s interesting about our results is the focus on a brief period exercise, which few investigators have examined. The point of the paper is to examine genes that were activated by exercise, not to claim that exercise is the only stimulus that can activate dentate granule cells. We can adjust the text to clarify this issue, but we see no point in additional experiments.

Additionally, Fos is universally accepted as a measure of neural activity, and we show definitively in Figure 1 that the single bout of exercise causes a 5-fold increase in Fos labeled cells within the tamoxifen-induced time window, and that the expression in homecage is very low given the sparse nature of the dentate gyrus. We confirm in vitro in Figure 1—figure supplement 1 that neurons in the Fos-TRAP mouse are similarly sensitive to depolarization as Fos immunohistochemistry. Thus our view is that this result is clear-cut and shows definitively that our stimulus (exercise) led to the Fos-TRAP labeled cells used for the analysis. We agree that a small fraction of Fos-TRAP labeled could have been activated within the taxomifen time window by a different stimulus, but that would not affect any of our conclusions. The suggestion to measure firing rate of the Fos-TRAP labeled cells is not a feasible experiment in vivo and would only show that Fos is activated by neural activity.

- Figure 2/Figure 6. All of the functional data involves comparison of TRAPed and non-TRAPed (control) GCs in the exercise condition. To confirm that the effects result from exercise, the author should address functional synaptic differences in TRAPed and control GCs in the non-exercise condition. This would address the possibility that TRAPed cells exhibit increased OML synaptic function in the absence of exercise.

As seen in Figure 2A, there is no difference in spine density between OML and MML in homecage Fos-TRAPed cells (i.e. in the absence of the exercise protocol). For the convenience of the reviewer we have included Author response image 1 with direct comparison of the data. Additionally, we have previously reported that OML and MML perforant path EPSCs in mature granule cells have similar amplitudes (Woods et al., 2018) using both electrical stimulation and pathway-specific Ch2R labeling of LEC and MEC inputs. Thus the appropriate control group that we used is neurons that were not Fos-labeled, but in the same slice, and in the case of the physiology experiments they are induced by the same exact stimulus in pairs of neighboring granule cells. We think this difficult experiment is unequivocal and the best possible control. Comparing between groups of animals, some exercised and some not, is not a good experimental design given the variability between slices in placement of stimulating electrodes, and thus would not be interpretable for the purpose of this manuscript.

**Author response image 1. respfig1:** Spine densities were similar in OML and MML of Fos-TRAPed homecage cells. (OML 3 days: 0.94 ± 0.1, MML: 0.99 ± 0.1, n=5).

- Figure 4 addresses the expression of Mtss1L in the DG in response to exercise, but the images are not very helpful. Higher magnification images are needed to illustrate the expression pattern. It is also unclear why the authors used the KOMP approach rather than the antibody shown in Figure 5A. Co-labeling of Mtss1L and Fos-TRAP before and after exercise should be obtained to confirm that it is upregulated only in Fos-TRAPed GCs.

The KOMP mice clearly show upregulation of MtsslL with exercise, which is the point of the figure. Figure 4 will be replaced, including higher magnification inserts allowing better visualization of the expression pattern in dentate granule cells. Unfortunately, none of the available antibodies against Mtss1L stain with high resolution in tissue and thus we used the galactosidase reporter expression of the KOMP mouse to examine expression in tissue. We acknowledge that this approach does not provide the cell compartment specificity that would be ideal. We are working on a KI reporter mouse but we do not think we should be required to create new antibodies or mice for this novel gene as these experiments extend beyond the scope of the manuscript and are not necessary for any of our conclusions.

- Results, fifth paragraph: Key reagent validation is necessary. The KOMP Mtss1L^tm1a(KOMP)Wtsi^ allele is reporter tagged allele with conditional potential. LacZ activity with this allele is not Cre dependent. LacZ activity can be eliminated with Flp expression. Treating with Cre will excise the Neo cassette and exons 7 and 8 leaving LacZ expressed and unaffected. Of concern is the lack of characterization of the allele and the apparently unnecessary manipulations on their animals (nuGFP-Cre injection). Additional controls need to be included that are required of those making or using an allele for the first time. The authors should more clearly explain the motivation for nuGFP-CRE line of experiments, as it is not clear in the text.

We agree with reviewer that LacZ activity is not Cre-dependent in the Mtss1L^tm1a(KOMP)Wtsi^ allele and we apologize for the misstatement that we will correct. Because the PGK promoter of the Neomycin gene is bidirectional, which may interfere with gene expression, KOMP recommends removing the Neo cassette Therefore, we did this in tissue-specific way using AAV nuGFP-Cre to remove PGK-Neo cassette in a tissue specific manner. Additional explanation will be added in the Materials and methods section of the manuscript for clarification.

We validated Mtss1L^-/-^ KO KOMP mice using RT-qPCR (see Author response image 2) and immunohistochemistry (see Author response image 2) in the cerebellum. Cerebellar Purkinje cells, which have high basal firing rate, interestingly, are the only cells in the brain with high basal Mtss1L expression (see Allen Brain Atlas).

**Author response image 2. respfig2:** Validation of Mtss1L^-/-^ KO KOMP mice. (**A**) RT-qPCR data confirmed Mtss1L deletion in Mtss1L^-/-^ KO KOMP mice in the cerebellum. Notably Mtss1L mRNA levels in the dentate gyrus were undetectable in wildtype (WT) and *Mtss1L*^-/-^ KO KOMP mice. mRNA was isolated from cerebellum and dentate gyrus tissue from 6 weeks old (WT) and *Mtss1L*^-/-^ KO KOMP mice. RT-qPCR data are from three animals per genotype, with each sample run in duplicate (p < 0.05, t-test). (**B**) Mtss1L immunohistochemistry was performed on WT and Mtss1L^-/-^ KO KOMP cerebellum. Purkinje cells were labeled with anti-Calbindin. Although quality of anti-Mtss1L is very low as seen in top panels, Mtss1L expression pattern in the cerebellum of WT mice overlapped with Calbindin^+^ Purkinje cells. Mtss1L expression was absent in the cerebellum of Mtss1L KO KOMP mice validating both the mouse line and the antibody.

- Additionally, several sequencing and array studies suggest Mtss1L expression in neurons is ambiguous, but high in radial glia (Saarikangas et al., 2008), astrocytes (astrocyternaseq.org), oligodendrocyte progenitors (https://web.stanford.edu/group/barres_lab/brain_rnaseq.html, described in Zhang et al., 2014, J. Neurosci.). Additional experiments to better characterize Mtss1L staining would be incredibly helpful in resolving this ambiguity.

Our results are not ambiguous with respect to expression in dentate granule cells as our experiments are focused on specific populations of activated neurons; and the RNA seq results were validated by PCR and with the in vitro experiments and KOMP beta-galactosidase reporter. We agree that a better antibody would be useful, but this is not possible until we have a KI reporter as antibodies with the necessary sensitivity are not available as shown above. However, the ambiguous results the reviewer/editor mentions can be explained by the activity-dependence of MtsslL and by the lack of cell- and condition-specific conditions in these prior studies. We can discuss this point more fully in the text. Of course, Mtss1L may have a role in other cell types such as glia, but that is beyond the scope of this manuscript.

- The Mtss1L paralog Mtss1 have multiple splice variants that are biologically important controlling localization in the cell and other functions. Given the similarity one may expect similar for Mtss1L and clearly documenting the spliceform used is essential. Related, Figure 5C and D – the interpretation of these overexpression experiments will depend strongly on the spliceform of Mtss1L used. Saarikangas et al. (2015) demonstrated the Mtss1 IMD domain alone is sufficient to induce extra spines, which is almost completely identical to Mtss1L. However, Mtss1 and Mtss1L have different overexpression phenotypes, suggesting domains outside the N-terminal IMD region are biologically important. Without explaining what form of Mtss1L is being used it's impossible for readers to determine whether these overexpression phenotypes reflect Mtss1L function, or whether a truncated form is mimicking roles established for Mtss1.

We should emphasize that the editor/reviewer is extrapolating from Mtss1, which is not the gene under study in this manuscript. The construct used for the overexpression studies was a full length Mtss1L

(https://www.ncbi.nlm.nih.gov/nuccore/BC060632.1) that includes the two known variants. The second variant differs in the 5' UTR, lacks a portion of the 5' coding region, and uses a downstream start codon compared to variant 1. It encodes isoform 2, which has a shorter N-terminus compared to isoform 1. The points raised about splice variants will be interesting to explore in the future, but we think the experiments are interpretable as presented. We can add to the Discussion that Mtss1 has splice variants and that this issue would be interesting to explore for Mtss1L.

- Figure 6D. The authors show that expressing Mtss1L is sufficient to alter spines in GCs in the absence of exercise, suggesting an important contribution to on-going synaptic plasticity. Thus, it seems counter-intuitive that knockdown of Mtss1L had no effect on EPSCs in the basal condition (Figure 6—figure supplement 2). This should be explained (as it relates to the first point).

We view this as the expected limitation of any over-expression experiment, particularly given that Mtss1L is not expressed in dentate granule cells in the basal condition. Thus the result is *not* counter-intuitive. As we show Mtss1L is not expressed at detectable levels in the absence of stimuli increasing neural activity – in our case a single bout of exercise. The value of the over-expression experiments is merely to show that Mtss1L is capable of increasing dendritic spines based on its function as an I-BAR protein. We can add to the text to make this clear.

[Editors’ notes: the authors’ response after being formally invited to submit a revised submission follows.]

Essential revisions:The reviewers were not fully satisfied about the revision response to point 2 relating to the relationship between the authors conclusions about Mtss1L-induced synapse/spine plasticity and exercise.There is no doubt that Fos+ cells represent an active population of GCs in the DG, sometimes referred to as "engram" cells. The Tonegawa lab has used a similar approach to show that Fos+ GCs have greater evoked EPSCs and spine density compared to Fos- GCs after a 5 min exposure to novel context (see Supplementary Figure 7 of Ryan et al., 2015). The authors should do a similar experiment – repeat the experiment of 2B in mice that have not undergone exercise – to test whether exercise is really required for this main outcome measure. The Ryan et al. data suggests that the synaptic function and spine difference between Fos+ and Fos- cells will exist in the absence of exercise (i.e. exercise is not required). The result of this experiment could differentiate whether Mtss1L is simply an activity-dependent gene downstream of Fos or whether it is specifically related to exercise-induced activity as indicated in the Abstract and Introduction. The outcome of this experiment does not change the mechanistic novelty of the role of Mtss1L in activity-dependent synaptic plasticity, but it could affect the conclusions that the authors make about this mechanism in the context of exercise. At the very least, the authors should discuss their results in the context of what has been reported about synapses/spines of Fos+ and Fos- GCs in the absence of exercise, which is very similar to what they are currently ascribing to an effect of exercise.It is unclear how closely the synaptic changes are associated with exercise rather than then Fos-expressing status of GCs (which could occur under various conditions and in the homecage). Since the authors strongly focus on exercise and subsequent cognitive benefits in the Introduction, it is important to confirm that these are exercise-mediated effects.

The point of our experiments was to use exercise as a physiologically relevant stimulus, and the experimental design was developed to allow us to examine the time course of activation of genes following a single bout of exercise. We used this stimulus because exercise is such an important physiological stimulus and relevant to humans. It is important to make clear that we did not suggest that exercise is the only stimulus that would activate granule cells. The editor/reviewer’s comment that “whether Mtss1L is simply an activity-dependent gene downstream of Fos…” understates our results as our experiments identified Mtss1L as an activity-dependent gene and effector of synaptic reorganization. Likewise, as we demonstrated in control experiments, Fos-TRAP was used as a proxy for neuronal activity and does not necessarily imply that Mtss1L activation is molecularly coupled to Fos, although it could be. We make no claims in that regard.

With regard to the Ryan et al. (2015), the authors did not address the laminar site of spines or EPSCs or attempt to link the stimulus to specific genes. Thus direct comparison is difficult, but we have no doubt that novel experiences will activate dentate granule cells. As a control for our exercise protocol, we have previously tested the effects of introducing a fixed running wheel into the environment for 2 hours – the equivalent of a novel object. Mice explored the wheel but did not run on it. This resulted in a modest increase in Fos-TRAPed cells (ca. 2-fold, vs. 4-fold for exercise) and Mtss1L expression (3-fold compared to 9-fold, by RT-PCR). This increase was above baseline but less than observed with our exercise protocol. This data makes the point that other stimuli can upregulate Mtss1L. We have included this data as a supplementary figure, and now make it clearer in the Discussion that we are not trying to argue (nor did the original version) that exercise is the only stimulus that might activate Mtss1L – in fact we fully expect it is not.